# Study of heavy metal resistance genes in *Escherichia coli* isolates from a marine ecosystem with a history of environmental pollution (arsenic, cadmium, copper, and mercury)

**Ashley S. Tseng**[1], **Marilyn C. Roberts**[2], **Scott J. Weissman**[3], **Peter M. Rabinowitz**[1,2,4]*

1 Department of Epidemiology, University of Washington, Seattle, Washington, United States of America,
2 Department of Environmental and Occupational Health Sciences, University of Washington, Seattle, Washington, United States of America, 3 Division of Infectious Diseases, Seattle Children's Hospital, Seattle, Washington, United States of America, 4 Center for One Health Research, Department of Environmental and Occupational Health Sciences, University of Washington, Seattle, Washington, United States of America

* peterr7@uw.edu

**Data Availability Statement:** All sequence data are available from the U.S. National Institutes of Health National Center for Biotechnology Information

## Abstract

We analyzed whole genome sequences of 308 *Escherichia coli* isolates from a marine ecosystem to determine the prevalence and relationships of heavy metal resistance genes (HMRGs) and antibiotic resistance genes (ARGs), as well as the presence of plasmid sequences. We screened all genomes for presence of 18 functional HMRGs conferring resistance to arsenic, cadmium, copper, or cadmium/mercury. In subset analyses, we examined geographic variations of HMRG carriage patterns in 224 isolates from water sources, and sought genetic linkages between HMRGs and ARGs in 25 genomes of isolates resistant to antibiotics. We found high carriage rates of HMRGs in all genomes, with 100% carrying at least one copy of 11 out of 18 HMRGs. A total of 173 (56%) of the isolates carried both HMRGs and plasmid sequences. In the 25 genomes of antibiotic-resistant isolates, 80% (n = 20) carried HMRGs, ARGs, and plasmid sequences, while 40% (n = 10) had linked HMRGs and ARGs on their assembled genomes. We found no evidence of geographic variation in HMRG frequency, nor any association between locational proximity to Superfund sites and co-carriage of HMRGs and ARGs. Our study findings indicate that HMRGs are common among *E. coli* in marine ecosystems, suggesting widespread heavy metal presence in water sources of a region with history of environmental pollution. Further research is needed to determine the role HMRGs play in driving antimicrobial resistance in human pathogens through genetic linkage and the value their detection in environmental bacterial genomes may offer as an indicator of environmental heavy metal pollution.

## Introduction

While there has been increasing scientific recognition of the global health threat posed by bacteria resistant to antibiotics, there has been less attention to the fact that many bacteria also

(NCBI) BioProject database (BioProject accession
number PRJNA283914). The NCBI Sequence Read
Archive IDs for the 308 included isolates were:
SRR11116088, SRR10947731, SRR10451710,
SRR11116091, SRR10451712, SRR10451715,
SRR10451717, SRR11251337, SRR10451719,
SRR12071682, SRR11116086, SRR10451708,
SRR10451725, SRR10536656, SRR10451709,
SRR10536658, SRR10536659, SRR10536660,
SRR10536661, SRR10536662, SRR10536663,
SRR10451711, SRR10536665, SRR10536666,
SRR10536667, SRR10536669, SRR10536670,
SRR10536671, SRR10536672, SRR10536673,
SRR10536674, SRR11536633, SRR11536634,
SRR10451718, SRR10546708, SRR10559557,
SRR11621679, SRR10600402, SRR11960525,
SRR10600523, SRR10451720, SRR11960529,
SRR10609374, SRR10609380, SRR10609382,
SRR10609383, SRR10609386, SRR10609387,
SRR10609388, SRR10609391, SRR10609392,
SRR10609393, SRR10613301, SRR10613302,
SRR11960626, SRR10947777, SRR10947785,
SRR10451721, SRR10947787, SRR10948144,
SRR10948120, SRR10948212, SRR10948217,
SRR10948273, SRR10948361, SRR11960630,
SRR10948365, SRR10948370, SRR10948371,
SRR11960632, SRR10948364, SRR11039029,
SRR11039030, SRR11059186, SRR11059188,
SRR11059190, SRR11059192, SRR11059194,
SRR11059196, SRR11067917, SRR11067918,
SRR11067919, SRR11068074, SRR11068075,
SRR11068076, SRR11068079, SRR11068080,
SRR11068085, SRR11068086, SRR11068087,
SRR11068088, SRR11068093, SRR11068159,
SRR11091346, SRR11091347, SRR11091348,
SRR11091349, SRR11091350, SRR11091351,
SRR11091352, SRR11091353, SRR11091354,
SRR11091355, SRR11091356, SRR11091357,
SRR11091358, SRR11091359, SRR11091361,
SRR11091364, SRR11113559, SRR11113562,
SRR11113564, SRR11113565, SRR11113566,
SRR11113567, SRR11113568, SRR11113569,
SRR11113570, SRR11113571, SRR11113572,
SRR11113573, SRR11113574, SRR11114574,
SRR11114576, SRR11114577, SRR11114578,
SRR11114579, SRR11114581, SRR11114627,
SRR11114628, SRR11114629, SRR12643357,
SRR11116073, SRR11116074, SRR11116076,
SRR11116083, SRR11116084, SRR11116085,
SRR11116104, SRR10959939, SRR11251334,
SRR11116089, SRR11116090, SRR11116092,
SRR11116072, SRR11116093, SRR11116094,
SRR11116095, SRR11116096, SRR11116097,
SRR11116098, SRR10451724, SRR11116100,
SRR11116101, SRR11116087, SRR11251328,
SRR11251336, SRR10600525, SRR10536657,
SRR11187233, SRR11187234, SRR11187235,

carry genes conferring resistance to heavy metal toxicity, and that these heavy metal resistance genes (HMRGs) may interact with genes conferring resistance to antibiotics. Evidence to support this comes from studies showing increased antimicrobial resistance in bacteria from environments contaminated with heavy metals, and a direct relationship between the carriage of HMRGs and the carriage of antibiotic resistance genes (ARGs) [1]. While the spread of ARGs among and between humans and animals is under intense study, less is known about the movement and sources of HMRGs in bacterial environmental reservoirs.

Although many heavy metals are found naturally in the environment, they can also be present at high concentrations as a result of human activities [2, 3]. Exposure to heavy metals at toxic levels can have detrimental effects on the health of humans and animals [2]. Commonly occurring toxic heavy metals relevant to public health include arsenic, cadmium, copper, lead, and mercury [4, 5]. Certain heavy metals, including copper, arsenic, mercury, and lead, have been used in the past as antimicrobial agents to treat and prevent bacterial infections, and some are still being used in agricultural settings today [6, 7]. Bacteria can acquire resistance to heavy metals as an evolutionary adaptation, which in turn may increase resistance to antibiotics [8]. There is a concern that antimicrobial resistance (AMR) may emerge in environments contaminated with heavy metals. There also may be a direct relationship between the carriage of HMRGs and the carriage of ARGs in common bacteria such as *Escherichia coli*. Consequently, efforts to address AMR may need to consider the role of heavy metals—in addition to the role of plasmids—in the movement of resistance genes through populations and ecosystems.

*E. coli* is a large and diverse species group of gram-negative bacteria which are found in the intestinal microbiome of humans and animals, as well as in the environment and as contamination in food products [9]. While most strains of *E. coli* are not harmful to humans, some *E. coli* strains are opportunistic pathogens which can cause diarrhea, urinary tract infections, respiratory illness and pneumonia, or other illnesses in humans and animals [9]. Because of its widespread nature in the gut of mammals, reptiles, wild avian species, in the environment, and in food, *E. coli* encounters diverse exposures and adapts to selective pressures [10].

Antibiotic resistance in *E. coli* is an increasing concern among public health practitioners due to its ability to accumulate genes conferring resistance to many clinically-relevant antimicrobial agents [11]. Studies have shown that there is often linkage between the carriage of ARGs and HMRGs on the same mobile elements [12–15]. A prior exploratory study characterized antibiotic resistance in *E. coli* samples from water and animal sources in the Salish Sea ecosystem, some of which were sampled from Superfund sites in the region [16]; however, heavy metal resistance was not previously characterized in these samples.

We analyzed a set of *E. coli* genomes of environmental and animal isolates sampled from a marine ecosystem with a history of environmental pollution in order to 1) characterize the presence of HMRGs, 2) determine if the carriage of HMRGs was associated with geographic location of the isolate, and 3) determine in a subset of isolates that were resistant to antibiotics if the carriage of HMRGs was associated with the carriage of ARGs.

## Materials and methods

### Data source

This study utilized whole genome sequencing (WGS) data from *E. coli* collected and isolated in a prior study of *E. coli* antibiotic resistance in the Salish Sea ecosystem of the Pacific Northwest region in the US [16]. The Salish Sea ecosystem, located between British Columbia in Canada and Washington (WA) in the US, contains a number of Superfund sites

SRR11187236, SRR11187237, SRR11187239, SRR12071684, SRR11187241, SRR11230643, SRR11230717, SRR11230720, SRR11230721, SRR11230722, SRR11230723, SRR11230724, SRR11237628, SRR11251324, SRR11251325, SRR11251326, SRR11251327, SRR10536664, SRR11251329, SRR11251330, SRR11251331, SRR10537325, SRR10537328, SRR11116105, SRR11251335, SRR10537329, SRR10559569, SRR11251338, SRR11251365, SRR11251366, SRR11251367, SRR11251368, SRR11536630, SRR11536631, SRR11116099, SRR10600403, SRR11536635, SRR11536636, SRR11536637, SRR11617886, SRR11617968, SRR11618034, SRR11618040, SRR11621676, SRR11621678, SRR11251332, SRR11621680, SRR11116102, SRR11960526, SRR11960527, SRR11960528, SRR11251333, SRR11960530, SRR11960617, SRR11960618, SRR11960622, SRR11960624, SRR11960625, SRR10959938, SRR11960628, SRR11960635, SRR11960631, SRR12643358, SRR10948215, SRR12071681, SRR11116103, SRR12071683, SRR10600524, SRR12174400, SRR12174404, SRR12174406, SRR12424844, SRR12424848, SRR12424849, SRR12424850, SRR12424853, SRR12424854, SRR12424855, SRR12424857, SRR12424858, SRR12424859, SRR12424861, SRR12584068, SRR12584069, SRR12584070, SRR12584071, SRR12584072, SRR12584073, SRR12584074, SRR12584075, SRR12584076, SRR12584077, SRR12584078, SRR12584079, SRR12584080, SRR12584081, SRR12584082, SRR12584083, SRR12584084, SRR12584085, SRR12584086, SRR12584089, SRR12586812, SRR12586813, SRR12586814, SRR12586815, SRR12586816, SRR12586819, SRR12586820, SRR12586821, SRR12586822, SRR12586823, SRR12586824, SRR12643343, SRR12643346, SRR12643347, SRR12643349, SRR12643353, SRR12643354, SRR12643356, SRR11187232, SRR11187240, SRR12643360, SRR12643361, SRR12643365, SRR12643367, SRR12643368, SRR12643371, SRR12663939, SRR12663940, SRR12663941, SRR12663942, SRR12663943, SRR12663944, SRR12706037, SRR12706039, SRR12706040, SRR12706041, SRR12706042, SRR12777348, SRR12777350, SRR12777351, SRR12824293, SRR11068078, SRR10546709, and SRR11230719.

**Funding:** This work was supported by the National Institute of Environmental Health Sciences at the National Institutes of Health (https://www.niehs. nih.gov/), grant number T32 ES015459 to A.S.T. The funders had no role in study design, data collection and analysis, decision to publish, or preparation of the manuscript.

designated by the US Environmental Protection Agency as polluted by hazardous waste, including heavy metals [17]. The prior study found varying levels of antibiotic resistance in 308 *E. coli* isolates from water and animal sources in the Salish Sea ecosystem [16]. A subset of isolates, 25 (8%) were phenotypically resistant to antibiotics and carried known antibiotic resistance genes [16].

## *E. coli* collection and isolation

The details of sampling have been previously described [16]. In brief, fecal samples from harbor seals (*Phoca vitulina*), harbor porpoises (*Phocoena phocoena*), river otters (*Lontra canadensis*), and English sole (*Parophrys vetulus*), and water samples from the Salish Sea watershed (fresh water from streams near marine beaches and marine water from four quadrants of the Salish Sea and locations near beaches) were collected during 2018–19 [16]. *E. coli* were isolated from these samples (N = 308) and tested for phenotypic and genotypic antibiotic susceptibility. A total of 224 of the *E. coli* isolates were from water samples and 84 isolates were obtained from samples of marine mammals. Fecal swabs were collected postmortem from harbor seals and harbor porpoises between Fall 2018 and Fall 2019. Detailed methods and isolates were described previously [18]. All but one live harbor seal fecal samples were collected by the WA Department of Fish and Wildlife staff from marina docks at various locations throughout the Puget Sound, where harbor seals defecate along the coast at haulout sites. The river otter fecal samples were collected from six of their latrine sites along the Duwamish River, Washington (May–September 2018) [16].

All the isolates sampled from the marine water, marine fish, and live seals were analyzed using broth dilution antibiotic susceptibility testing with the SensititreTM Nephelometer (Thermo Fisher Scientific, Waltham, MA, USA) using the manufacturer's instructions at the WA State Department of Health's laboratory. The panels were read using the approved Sensititre SWIN software and were also inspected visually by technicians for microbial growth. The minimum inhibitory concentration for each antibiotic in mg/mL was determined using the Clinical and Laboratory Standards Institute interpretive criteria [19]. This allowed each isolate's susceptibility pattern (susceptible, intermediate resistant, or resistant) to be assigned to the following antibiotics: amikacin, aztreonam, cefepime, cefotaxime, ceftazidime, ciprofloxacin, doripenem, doxycycline, ertapenem, gentamicin, imipenem, levofloxacin, meropenem, minocycline, piperacillin/tazobactam, ticarcillin/clavulanic acid, tigecycline, tobramycin, and trimethoprim/sulfamethoxazole. *E. coli* from dead seals and porpoises were tested using the bioMérieux VIETK instrument (Durham, North Carolina, US). The *E. coli* isolated from river otters, fresh water, marine water by beaches, and a rescued seal pup were tested using a standard disk diffusion assay according to the Clinical and Laboratory Standards Institute [19]. Standard *E. coli* negative and positive controls were included in each assay.

All 308 isolates were forwarded by the WA State Department of Health's Antimicrobial Resistance Laboratory to the GenomeTrakr program of the Food and Drug Administration for WGS performed using Illumina (Illumina, San Diego, California, US). The sequence data were deposited with links to BioProject accession number PRJNA283914 in the US National Institutes of Health National Center for Biotechnology Information (NCBI) BioProject database (https://www.ncbi.nlm.nih.gov/bioproject/). The NCBI Sequence Read Archive (https://www. ncbi.nlm.nih.gov/sra/) IDs for the 308 included isolates were: SRR11116088, SRR10947731, SRR10451710, SRR11116091, SRR10451712, SRR10451715, SRR10451717, SRR11251337, SRR10451719, SRR12071682, SRR11116086, SRR10451708, SRR10451725, SRR10536656, SRR10451709, SRR10536658, SRR10536659, SRR10536660, SRR10536661, SRR10536662,

**Competing interests:** The authors have declared that no competing interests exist.

SRR10536663, SRR10451711, SRR10536665, SRR10536666, SRR10536667, SRR10536669, SRR10536670, SRR10536671, SRR10536672, SRR10536673, SRR10536674, SRR11536633, SRR11536634, SRR10451718, SRR10546708, SRR10559557, SRR11621679, SRR10600402, SRR11960525, SRR10600523, SRR10451720, SRR11960529, SRR10609374, SRR10609380, SRR10609382, SRR10609383, SRR10609386, SRR10609387, SRR10609388, SRR10609391, SRR10609392, SRR10609393, SRR10613301, SRR10613302, SRR11960626, SRR10947777, SRR10947785, SRR10451721, SRR10947787, SRR10948144, SRR10948120, SRR10948212, SRR10948217, SRR10948273, SRR10948361, SRR11960630, SRR10948365, SRR10948370, SRR10948371, SRR11960632, SRR10948364, SRR11039029, SRR11039030, SRR11059186, SRR11059188, SRR11059190, SRR11059192, SRR11059194, SRR11059196, SRR11067917, SRR11067918, SRR11067919, SRR11068074, SRR11068075, SRR11068076, SRR11068079, SRR11068080, SRR11068085, SRR11068086, SRR11068087, SRR11068088, SRR11068093, SRR11068159, SRR11091346, SRR11091347, SRR11091348, SRR11091349, SRR11091350, SRR11091351, SRR11091352, SRR11091353, SRR11091354, SRR11091355, SRR11091356, SRR11091357, SRR11091358, SRR11091359, SRR11091361, SRR11091364, SRR11113559, SRR11113562, SRR11113564, SRR11113565, SRR11113566, SRR11113567, SRR11113568, SRR11113569, SRR11113570, SRR11113571, SRR11113572, SRR11113573, SRR11113574, SRR11114574, SRR11114576, SRR11114577, SRR11114578, SRR11114579, SRR11114581, SRR11114627, SRR11114628, SRR11114629, SRR12643357, SRR11116073, SRR11116074, SRR11116076, SRR11116083, SRR11116084, SRR11116085, SRR11116104, SRR10959939, SRR11251334, SRR11116089, SRR11116090, SRR11116092, SRR11116072, SRR11116093, SRR11116094, SRR11116095, SRR11116096, SRR11116097, SRR11116098, SRR10451724, SRR11116100, SRR11116101, SRR11116087, SRR11251328, SRR11251336, SRR10600525, SRR10536657, SRR11187233, SRR11187234, SRR11187235, SRR11187236, SRR11187237, SRR11187239, SRR12071684, SRR11187241, SRR11230643, SRR11230717, SRR11230720, SRR11230721, SRR11230722, SRR11230723, SRR11230724, SRR11237628, SRR11251324, SRR11251325, SRR11251326, SRR11251327, SRR10536664, SRR11251329, SRR11251330, SRR11251331, SRR10537325, SRR10537328, SRR11116105, SRR11251335, SRR10537329, SRR10559569, SRR11251338, SRR11251365, SRR11251366, SRR11251367, SRR11251368, SRR11536630, SRR11536631, SRR11116099, SRR10600403, SRR11536635, SRR11536636, SRR11536637, SRR11617886, SRR11617968, SRR11618034, SRR11618040, SRR11621676, SRR11621678, SRR11251332, SRR11621680, SRR11116102, SRR11960526, SRR11960527, SRR11960528, SRR11251333, SRR11960530, SRR11960617, SRR11960618, SRR11960622, SRR11960624, SRR11960625, SRR10959938, SRR11960628, SRR11960635, SRR11960631, SRR12643358, SRR10948215, SRR12071681, SRR11116103, SRR12071683, SRR10600524, SRR12174400, SRR12174404, SRR12174406, SRR12424844, SRR12424848, SRR12424849, SRR12424850, SRR12424853, SRR12424854, SRR12424855, SRR12424857, SRR12424858, SRR12424859, SRR12424861, SRR12584068, SRR12584069, SRR12584070, SRR12584071, SRR12584072, SRR12584073, SRR12584074, SRR12584075, SRR12584076, SRR12584077, SRR12584078, SRR12584079, SRR12584080, SRR12584081, SRR12584082, SRR12584083, SRR12584084, SRR12584085, SRR12584086, SRR12584089, SRR12586812, SRR12586813, SRR12586814, SRR12586815, SRR12586816, SRR12586819, SRR12586820, SRR12586821, SRR12586822, SRR12586823, SRR12586824, SRR12643343, SRR12643346, SRR12643347, SRR12643349, SRR12643353, SRR12643354, SRR12643356, SRR11187232, SRR11187240, SRR12643360, SRR12643361, SRR12643365, SRR12643367, SRR12643368, SRR12643371, SRR12663939, SRR12663940, SRR12663941, SRR12663942, SRR12663943, SRR12663944, SRR12706037, SRR12706039, SRR12706040, SRR12706041, SRR12706042, SRR12777348, SRR12777350, SRR12777351, SRR12824293, SRR11068078, SRR10546709, and SRR11230719.

## Selection of heavy metals of interest

Informed by a systematic literature review of HMRGs in *E. coli*, we selected five heavy metals of public health concern [4] with confirmed presence in the Salish Sea region [20, 21]: arsenic, cadmium, copper, lead, and mercury.

## Screening of *E. coli* genomes for heavy metal resistance genes

We constructed a novel bioinformatic pipeline based on existing bioinformatic tools to analyze both HMRGs and ARGs.

**Sequence retrieval, filtering, and assembly.** First, the sequences were retrieved from the 308 isolates from the National Center for Biotechnology Information Sequence Read Archive (SRA) using SRA Toolkit [22] as FASTQ files. Then, the raw paired-end sequence data were filtered using Trimmomatic [23] with a sliding window trimming size of 4 bases and required average quality of 20, minimum read length of 25 bases, and removal of Illumina adapters with a maximum seed mismatch count of 2, palindrome clip threshold of 40, and simple clip threshold of 15. Next, genomes were assembled for each sample (contigs) using MEGAHIT with default settings [24]. The open reading frames were identified with Prodigal [25] for putative genes using default settings.

**Screening for heavy metal resistance genes.** Using the BacMet-Scan command line tool, we cross-referenced the translated proteins to the antibacterial biocide and metal resistance genes (BacMet) database [26] with a DIAMOND Basic Local Alignment Search Tool for Proteins (BLASTP) search to identify HMRGs for each sequence.

BacMet-Scan screens for a total of 171 HMRGs in its experimentally confirmed resistance gene database (version 2.0) [26]. We used the experimentally confirmed database from BacMet as opposed to the predicted resistance genes database to allow for comparison between the current study results and prior findings in the literature. Of 171 possible HMRGs in the BacMet database, we screened for HMRGS for five heavy metals (arsenic, cadmium, copper, lead, and mercury), which are often associated with mobile elements [27]. A total of 22 HMRGs were selected: arsenic (*arsB*, *arsC*, *arsD*, *arsR*), cadmium/mercury (*dsbA*, *dsbB*, *robA*), cadmium (*ygiW*, *zinT/yodA*), copper (*bhsA/ycfR/comC*, *comR/ycfQ*, *cusR/ylcA*, *cusS*, *cutA*, *cutC*, *cutF/nlpE*, *dsbC*, *pcoA*, *pcoB*, *pcoC*, *pcoD*). For our analyses, *pcoA*, *pcoB*, *pcoC*, and *pcoD* were consolidated to a singular HMRG (*pcoABCD*) since they are each required for the copper-inducible expression of copper resistance and part of the copper resistance determinant system [26].

There were 18 HMRGs on our final screening list: four functional arsenic resistance genes, three cadmium/mercury resistance genes, two cadmium genes, and nine copper genes. We used a 90% identity match threshold in the alignment between the best matching HMRG in BacMet and the corresponding sequence in the input genome and 100% minimum length (i.e., the alignment must cover the entire length of the HMRG in the database with no gap openings). There were no functional lead resistance genes that met our inclusion criteria.

Among the isolates collected from water sources, we additionally assessed if the occurrence of HMRGs varied by geographic location in the Salish Sea ecosystem.

## Comparison to genotypic and phenotypic antibiotic resistance

We defined co-resistance as carriage of at least one HMRG and at least one ARG by an *E. coli* isolate. To explore genetic linkage between HMRGs and ARGs, we performed a sub-analysis on the 25 *E. coli* isolates that had previously been found to carry 1–4 ARGs [16]. Twenty-three were phenotypically antibiotic resistant and two isolates were tetracycline-intermediate resistant but carried known *tet* genes *tet*(A) or *tet*(B). We defined linkage as the physical location of

a HMRG and an ARG next to one another in the same DNA sequence, whether chromosome or plasmid [28].

Among the 224 isolates from water sources, we assessed if the carriage of HMRGs differed by proximity to Superfund sites or geographic location. We used the Public Health Agency of Canada's Staramr tool [29] to scan *E. coli* genome contigs against the Center for Genomic Epidemiology's ResFinder (2022-05-24 update) [30–32] and PointFinder (2021-02-01 update) [33] databases using the default 90% identity match and 60% minimum length coverage (percent overlap) of the target gene thresholds (as suggested by the Center for Genomic Epidemiology [31]), as well as 100% minimum total length coverage to be comparable with past analyses of the isolates [16]. To identify plasmid sequences among the *E. coli* isolate genomes, we scanned contigs against PlasmidFinder (2021-11-29 update) (0.8.0.dev2) [34, 35] using the same criteria.

To determine if there were significant associations between carriage of ARGs and the HMRGs that were not carried by all isolates, we ran Pearson's Chi-squared (two-sided) test with a Yates' correction for continuity due to small numbers at the $p<0.10$ and $p<0.05$ levels.

## Results

### Carriage of heavy metal resistance genes

At least one HMRG of interest was detected in every study isolate. Each isolate carried a range of 14–20 copies of HMRGs of interest, with some isolates carrying multiple copies of the same gene (Table 1). Eleven (61%) of the 18 HMRGs screened were identified in all 308 isolates: *arsC*, *bhsA/ycfR/comC*, *comR/ycfQ*, *cutA*, *cutC*, *cutF/nlpE*, *dsbA*, *dsbB*, *dsbC*, *robA*, and *ygiW* (S1 Table). The copper resistance gene *cutF/nlpE* was the most prevalent HMRG in our data, identified 321 times, which included 10 isolates each carrying two copies of the gene and one isolate carrying four copies (Fig 1). A cadmium/mercury resistance gene, *robA*, was the second most prevalent HMRG (n = 311), also carried by all study isolates, with one isolate from a river otter in the Green River Natural Area carrying three copies. The third most prevalent HMRG (n = 310) was *arsC*, an arsenic resistance gene, with two isolates from water sources (South Salish Sea and Strait of Juan de Fuca) carrying two copies each.

In contrast, the copper resistance gene *pcoABCD* was identified only seven times in the samples, primarily in *E. coli* isolates from mammals (river otters and harbor seals), and one isolate from water. We found an association between the presence of this gene and the presence of certain ARGs. Specifically, six ARGs were found to be significantly associated with *pcoABCD* at the $p<0.10$ level: *aac(3)-IV*, *aph(3')-Ia*, *aph(4)-Ia*, *aph(6)-Id*, *lnu*(F), and *tet*(B); however, the isolates carrying the aminoglycoside genes were previously shown not to be phenotypically resistant [16] (S2 Table). An arsenic gene, *arsD*, was identified in 73% (n = 11) of the isolates from the South Salish Sea, Strait of Juan de Fuca, and North Salish Sea, with two isolates from river otters in Hamm Creek and one isolate from a harbor seal in Fort Flagler Historical State Park. The fluoroquinolone resistance gene *qnrB82*, carried by only one out of 25 isolates, was found to be significantly associated with *arsD* at the $p<0.05$ level, though the isolate was not ciprofloxacin-resistant [16]. No other HMRG-ARG associations were statistically significant.

### Geographic variation of heavy metal resistance genes in isolates from water sources

Of the 18 HMRGs that we screened, only seven HMRGs were not universally found in all study isolates: *arsB*, *arsD*, *arsR*, *zinT/yodA*, *cusR/ylcA*, *cusS*, and *pcoABCD*. Nearly all (≥95%)

**Table 1. Count of selected heavy metal resistance genes (HMRGs) in 308 *Escherichia coli* isolates by Salish Sea region or source.**

| Isolate Source | Arsenic | | | | Cadmium/ Mercury | | | Cadmium | | Copper | | | | | | | | | Total HMRGs Count[b] |
|---|---|---|---|---|---|---|---|---|---|---|---|---|---|---|---|---|---|---|---|
| | *arsB*[a] | *arsC* | *arsD*[a] | *arsR*[a] | *dsbA* | *dsbB* | *robA* | *ygiW* | *zinT/ yodA*[a] | *bhsA/ ycfR/ comC* | *comR/ ycfQ* | *cusR/ ylcA*[a] | *cusS*[a] | *cutA* | *cutC* | *cutF/ nlpE* | *dsbC* | *pcoABCD*[a] | |
| **Water** | | | | | | | | | | | | | | | | | | | |
| North Salish Sea (n = 52) | 45 | 52 | 2 | 44 | 52 | 52 | 52 | 52 | 52 | 52 | 52 | 52 | 52 | 52 | 52 | 56 | 52 | 0 | 823 |
| Central Salish Sea (n = 55) | 41 | 55 | 0 | 40 | 55 | 55 | 56 | 55 | 55 | 55 | 55 | 55 | 56 | 55 | 55 | 59 | 55 | 0 | 857 |
| South Salish Sea (n = 57) | 46 | 58 | 3 | 46 | 58 | 57 | 57 | 57 | 54 | 57 | 57 | 56 | 56 | 57 | 57 | 58 | 58 | 0 | 892 |
| Strait of Juan de Fuca (n = 52) | 38 | 53 | 3 | 38 | 52 | 52 | 52 | 52 | 52 | 52 | 52 | 51 | 52 | 52 | 52 | 53 | 52 | 1 | 809 |
| Fresh Water (n = 5) | 3 | 5 | 0 | 3 | 5 | 5 | 5 | 5 | 5 | 5 | 5 | 5 | 5 | 5 | 5 | 5 | 5 | 0 | 76 |
| Marine Water by Beaches (n = 3) | 3 | 3 | 0 | 3 | 3 | 3 | 3 | 3 | 3 | 3 | 3 | 3 | 3 | 3 | 3 | 3 | 3 | 0 | 48 |
| **Marine Mammals** | | | | | | | | | | | | | | | | | | | |
| Harbor Seals (n = 51) | 31 | 51 | 1 | 31 | 51 | 52 | 51 | 51 | 51 | 51 | 51 | 51 | 51 | 51 | 51 | 53 | 51 | 3 | 783 |
| Harbor Porpoises (n = 7) | 3 | 7 | 0 | 3 | 7 | 7 | 7 | 7 | 7 | 7 | 7 | 7 | 7 | 7 | 7 | 7 | 7 | 0 | 104 |
| River Otters (n = 24) | 23 | 24 | 2 | 22 | 24 | 24 | 26 | 24 | 24 | 24 | 24 | 24 | 24 | 25 | 24 | 25 | 24 | 3 | 390 |
| Sole (n = 2) | 2 | 2 | 0 | 2 | 2 | 2 | 2 | 2 | 2 | 2 | 2 | 2 | 2 | 2 | 2 | 2 | 2 | 0 | 32 |
| **Total (N = 308)** | 235 | 310 | 11 | 232 | 309 | 309 | 311 | 308 | 305 | 308 | 308 | 306 | 308 | 309 | 308 | 321 | 309 | 7 | — |

[a]This HMRG was not found in every study isolate and was only carried by some isolates. *arsB* was carried by 231 isolates; *arsD* was carried by 11 isolates; *arsR* was carried by 230 isolates; *zinT/yodA* was carried by 305 isolates; *cusR/ylcA* was carried by 306 isolates; *cusS* was carried by 306 isolates; and *pcoABCD* was carried by 7 isolates.

[b]If the count for a HMRG is 3 in the table, it means that there are 3 of the same HMRG identified in isolates from a given source (e.g., South Salish Sea).

of the isolates from water sources carried copies of *zinT/yodA*, *cusR/ylcA*, and *cusS* (Table 2). In contrast, carriage of *arsD* and *pcoABCD* were uncommon in our study isolates (8/224 or 3.6% and 1/224 or 0.4%, respectively).

Seven HMRGs were not ubiquitous in terms of their distribution across four quadrants of the Salish Sea (North Salish Sea; Central Salish Sea, with samples from the fresh water and marine water by beaches; South Salish Sea; and Strait of Juan de Fuca). Of the seven non-universally found HMRGs, *arsB* and *arsR* were the most prevalent among the isolates from water samples in the four different quadrants, with 100% of these isolates carrying both genes. We additionally compared the HMRGs carried by isolates also carrying expressed ARGs (n = 25) with those carried by isolates without ARGs (n = 276; S1 File).

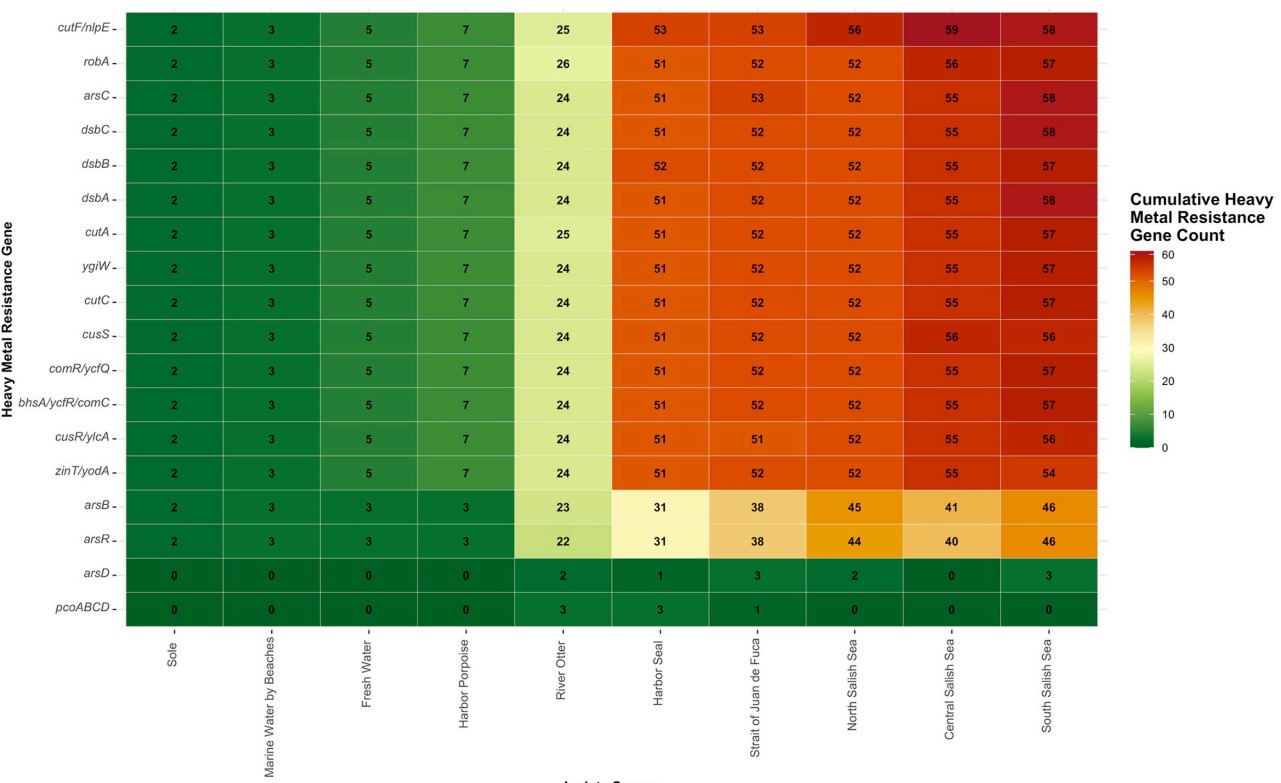

**Fig 1. Total count of heavy metal resistance genes per *E. coli* isolate source (N = 308).** Number of isolates per source: North Puget Sound (n = 52), Central Puget Sound (n = 55), South Puget Sound (n = 57), Strait of Juan de Fuca (n = 52), Fresh Water (n = 5), Marine Water by Beaches (n = 3), Harbor Seal (n = 51), Harbor Porpoise (n = 7), River Otter (n = 24), Sole (n = 2).

Fifteen percent (n = 45) of the *E. coli* isolates in the study were from sources located near (i.e., directly within or geographically adjacent to) Superfund sites currently on the US Environmental Protection Agency's National Priorities List. Of those 45 isolates, there were seven isolates from water samples in the Central Salish Sea near the Naval Undersea Warfare Engineering Station, Puget Sound Naval Shipyard, and Old Navy Dump/Manchester Laboratory Superfund sites (S1 Table). These seven isolates did not carry more total copies of HMRGs (range per isolate: 14–17) than the other 48 isolates collected from non-Superfund adjacent locations in the Central Salish Sea (range: 14–17). None of the isolates from water samples

**Table 2. Number of non-universal heavy metal resistance genes (HMRGs) carried by *Escherichia coli* isolates of water sources in the Salish Sea (N = 224).**

| Salish Sea Quadrant | Arsenic | | | Cadmium | Copper | | | Total Isolates Per Region |
|---|---|---|---|---|---|---|---|---|
| | *arsB* | *arsD* | *arsR* | *zinT/yodA* | *cusR/ylcA* | *cusS* | *pcoABCD* | |
| | n (%) | | | | | | | |
| North Salish Sea | 44 (85) | 2 (4) | 44 (85) | 52 (100) | 52 (100) | 52 (100) | 0 (0) | 52 |
| Central Salish Sea | 40 (73) | 0 (0) | 40 (73) | 55 (100) | 55 (100) | 55 (100) | 0 (0) | 55 |
| Fresh Water | 3 (60) | 0 (0) | 3 (60) | 5 (100) | 5 (100) | 5 (100) | 0 (0) | 5 |
| Marine Water by Beaches | 3 (100) | 0 (0) | 3 (100) | 3 (100) | 3 (100) | 3 (100) | 0 (0) | 3 |
| South Salish Sea | 46 (81) | 3 (5) | 46 (81) | 54 (95) | 56 (98) | 56 (98) | 0 (0) | 57 |
| Strait of Juan de Fuca | 37 (71) | 3 (6) | 37 (71) | 52 (100) | 51 (98) | 51 (98) | 1 (2) | 52 |

collected in the North Salish Sea, South Salish Sea, Strait of Juan de Fuca, fresh water, or marine water by beaches were near Superfund sites on the National Priorities List, however, all exhibited universal carriage of HMRGs (range: 14–20).

## Carriage of plasmids

Overall, 173 (56%) of the study isolates carried plasmid sequences. Out of 25 isolates with phenotypic resistance to antibiotics and carriage of at least one ARG, 20 (80%) also carried at least one plasmid sequence. Of the 283 isolates with no ARGs detected by WGS, 152 (54%) carried plasmid sequences (S3 Table). Four of these isolates (CWG7G, CWG7H, GRNRA4F, and 342381-006-850) were phenotypically resistant to antibiotics and carried plasmid sequences, but no ARGs were detected by WGS (S3 Table) [16].

## Antibiotic resistance genes, heavy metal resistance genes, plasmids, and Superfund sites

The 25 isolates that carried ARGs and that were phenotypically non-susceptible (resistant or intermediate susceptible) to antibiotics included two isolates from the Central Salish Sea, three isolates from fresh water, eight isolates from harbor seals, four isolates from the North Salish Sea, and eight isolates from river otters (Table 2). Overall, the most common ARGs reported were *tet*(B) and *aph(6)-Id* (Fig 2 and S1 Fig). We observed that 40% (n = 10) of these non-

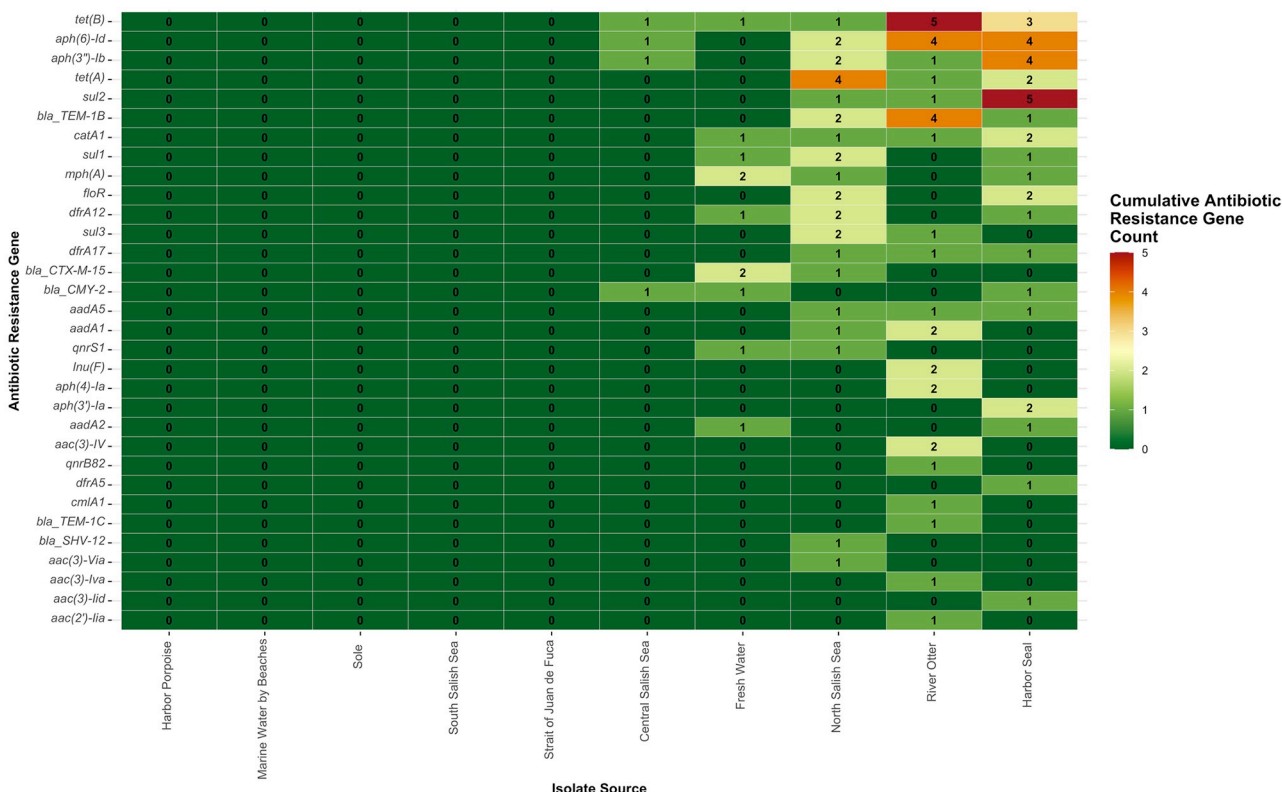

**Fig 2. Total count of antibiotic resistance genes per *E. coli* isolate source (N = 308).** Number of isolates per source: North Puget Sound (n = 52), Central Puget Sound (n = 55), South Puget Sound (n = 57), Strait of Juan de Fuca (n = 52), Fresh Water (n = 5), Marine Water by Beaches (n = 3), Harbor Seal (n = 51), Harbor Porpoise (n = 7), River Otter (n = 24), Sole (n = 2).

susceptible isolates had HMRGs and non-functional ARGs that were located next to each other and assumed to be linked on their assembled genomes (chromosome or plasmid sequences).

Among the subset of isolates with phenotypic resistance (n = 25), the ARGs *catA1* and *sul2* were found with the HMRG *cusS*, both located on the IncQ1 plasmid sequence in two harbor seal isolates, suggesting potential co-selection for *catA1*, *sul2*, and *cusS* resistance genes on the IncQ1 plasmid (using 90% identity match and 60% minimum length thresholds). IncQ1 plasmids have been shown to carry *sul2*, *str*AB, and *tet*(A) genes [36]. In our study, though found in both water and animal isolates, *sul2* was carried primarily in isolates from marine mammals (five harbor seals and one river otter). Further, we found that *sul2* was linked with HMRGs in three of the marine mammals isolates and one North Salish Sea water isolate.

**E. coli isolates from water sources.**    In the two isolates taken from water samples in Drayton Harbor, North Salish Sea (339942-001-501 and 339942-002-506), the position of the *tet*(A) gene was linked with all identified HMRGs except for *bhsA/ycfR/comC*, a copper resistance gene (Table 2). Further, in both of those isolates, the *qnrB19* ARG was linked with arsenic and copper resistance genes (*arsB* and *cusS*, respectively), in the chromosome. While both isolates carried the IncX1 plasmid sequence, and the isolate 339942-002-506 also carried the IncFIB (AP001918) plasmid sequence and demonstrated intermediate phenotypic resistance to ciprofloxacin, no ARGs or HMRGs were located on the plasmid sequences in either isolate. Additional findings of *E. coli* isolates from water sources are reported in the S1 File.

**E. coli isolates from marine mammal sources.**    Among the eight phenotypically antibiotic-resistant *E. coli* isolated from harbor seals, the most common ARG-HMRG combinations consisted of the *sul2* gene and the following 14 HMRGs: *arsC*, *bhsA/ycfR/comC*, *comR/ycfQ*, *cusR/ylcA*, *cusS*, *cutA*, *cutC*, *cutF/nlpE*, *dsbA*, *dsbB*, *dsbC*, *robA*, *ygiW*, and *zinT/yodA*. The *sul2* gene is often associated with integrons (S2 Fig). Three of the isolates from harbor seals were phenotypically resistant to doxycycline and three were phenotypically resistant to trimethoprim/sulfamethoxazole. One harbor seal (EPA Dock G Cip 1#5) carried *aph(3″)-Ib*, *aph(6)-Id*, *catA1*, *floR*, *sul2*, and *tet*(A) ARGs, one copy of each HMRG except for *arsD* and *pcoABCD*, and was phenotypically resistant to ciprofloxacin, levofloxacin, and tetracycline. For the isolate EPA Dock G Cip 1#5, the positions of the *catA1* and *sul2* genes were upstream from the position of the *cusS* copper resistance gene and was associated with the IncQ1 plasmid sequence. For one of the isolates taken from a harbor seal in Seattle (SKMMR2020-01-025 Fecal #1), the positions of the *catA1* and *sul2* ARGs were located upstream of the *cusS* copper resistance gene and associated with the IncQ1 plasmid.

Among the eight *E. coli* isolates from river otters, all carried 14–17 HMRGs. The most common ARG-HMRG combination was between *tet*(B) and *cutF/nlpE*, which occurred in five (62%) of those isolates (project IDs: BR1E, BR1F, CWG3I, HAM5E, HAM6D) (S3 Fig). In an isolate from a river otter in Hamm Creek (HAM5E), the positions of ARGs *qnrB19*, *aph(6)-Id*, and *catA1* overlapped with the position of nine HMRGs (*arsB*, *comR/ycfQ*, *cusS*, *cutC*, *cutF/nlpE*, *dsbA*, *dsbC*, *robA*, *zinT/yodA*) and were physically located on the same Col(pHAD28) plasmid sequence. The isolate HAM5E was phenotypically resistant to ampicillin, trimethoprim/sulfamethoxazole, tetracycline, minocycline, and sulfisoxazole. Additional results are reported in the S1 File.

## Discussion

This study found high carriage rates of HMRGs for arsenic, cadmium, copper, and mercury in all genomes of *E. coli* isolates collected from the Salish Sea marine environment. Among the isolates with phenotypic resistance to antibiotics as well as ARGs, we identified linkage

between HMRGs and ARGs. While only a few of the isolates in the study had HMRGs and ARGs linked to plasmid sequences, there was greater prevalence of plasmid sequences among ARG-carrying isolates than among non-ARG carrying isolates. Finally, we did not find measurable differences in the total number of HMRG copies carried by isolates from water sources according to geographic location or proximity to known Superfund sites.

Overall, there was a high prevalence of HMRGs across arsenic, cadmium, copper, and mercury functional resistance genes observed in our study isolates, suggesting prominent and universal exposure to heavy metals in the environment and among marine mammals in the Salish Sea. The most prevalent HMRGs identified in the isolates were a copper resistance gene *cutF/nlpE*, a cadmium/mercury resistance gene *robA*, and an arsenic resistance gene *arsC*.

In general, there was a high prevalence of copper resistance genes in *E. coli* isolates from the various water sources and marine mammals. The copper homeostasis protein encoded by *cutF* could be involved in both copper efflux and the delivery of copper to copper-dependent enzymes [26]. When overproduced, the *cutF/nlpE* gene product induces DegP, which acts as a protease in the acid resistance of *E. coli* [37] and is required for bacterial survival in harsh environmental conditions, including high temperatures [38]. Our finding of copper resistance genes in *E. coli* isolated throughout the Salish Sea ecosystem suggests widespread bacterial exposure to copper that should be further investigated.

The second most prevalent HMRG we identified was the *robA* gene, encoding a transcriptional activator that confers bacterial resistance to both cadmium and mercury resistance [26]. When overexpressed, the *robA* gene product induces multiple antibiotic and heavy metal resistance traits, as well as organic solvent tolerance in *E. coli* [39, 40]. In our study, five isolates that carried *robA* were phenotypically resistant to tetracycline, one was phenotypically resistant to chloramphenicol, and one intermediate-susceptible to kanamycin.

The third most prevalent HMRG found was an arsenic resistance gene, *arsC*, with the function of reducing arsenate to the more-toxic arsenite [26]. Having multiple copies of the *ars* operon increases resistance to arsenate. Prior studies have shown the widespread prevalence of *arsC* genes in water sources globally [15, 41, 42], suggesting high levels of arsenic pollution in the environment. Researchers have recommended bacteria carrying *arsC* as a potential bioremediation strategy in areas heavily contaminated with arsenic [43]. Examining the isolates from water sources only, *arsB* and *arsR* were the most prevalent HMRGs—a finding which is consistent with a recent study of *Staphylococcus aureus* in macaques, which found *arsB* and *arsR* in all study strains [44].

In a subset of 25 isolates that were previously found to be resistant to antibiotics [16], we observed overall high rates of co-resistance of HMRG and ARGs, and even genetic linkage of certain HMRGs and ARGs in a minority of isolates. The resistome (i.e., collection of all ARGs) [45] in the environment has expanded due to resistant organisms in human and environmental waste entering the environment without adequate treatment, in addition to the possibility of antibiotic residues and heavy metal pollution. *E. coli* isolates from human, animal, and environmental sources globally have revealed the diversity of ARGs from the same geographical reservoir that can be accumulated, carried, and transferred [16, 46, 47]. Further, previous work has shown linkages between selection for ARGs and HMRGs, specifically through co-occurrence [1] and cross-selection, in livestock [12, 13, 48], sediments [49], soils [50–52], wastewater treatment plants [53, 54], and in various water sources [15, 55–57]. The interactions of human, animal, and environmental factors in bacterial antimicrobial resistance call for an integrated approach to understanding transmission and evolution of heavy metals and antimicrobial resistance.

Due to their remarkable capacity to carry aggregates of ARGs, HMRGs, and virulence factors, plasmids spread drug resistance throughout bacterial populations via transduction or

conjugation [27]. Plasmids may play a potentially important role in the movement of resistance genes in the Salish Sea ecosystem: more than half of the sequences in our study carried both HMRGs and plasmids, and a greater proportion of the isolates carrying ARGs carried plasmids compared to isolates without ARGs. The *sul2* gene is often associated with small non-conjugative plasmids [58] or on large transmissible multi-resistance plasmids [59], such as the IncQ1 plasmid. While the acquisition of multiple sulfonamide resistance genes has been reported previously [60], in our study, we only had one isolate (from Portage Bay) that carried copies of both *sul1* and *sul2* genes, and only the *sul2* and *catA1* genes were located on the IncQ1 plasmid. A study of *E. coli* isolates from domestic animals and children identified carriage of both the *catA1* gene and the IncQ1 plasmid in the same isolate, however, the association of *catA1* with IncQ1 was not specifically examined [61]. Although we identified potential linkage for *catA1*, *sul2*, and *cusS* resistance genes on the IncQ1 plasmid for two harbor seal isolates in our study, no other studies have, to our knowledge, found *cusS* to be located with IncQ1.

While there is a history of hazardous waste pollution in the Salish Sea, we did not observe differences in carriage of HMRGs in *E. coli* isolates by proximity to Superfund sites. Further, we found that HMRGs were more common than ARGs in *E. coli* isolates, suggesting that exposures to heavy metals may be greater than exposure to antibiotics in this ecosystem. This may be because unlike antibiotics, heavy metals are stable and cannot be eliminated easily from the environment [62]. Thus, they persist in the environment much longer than do active antibiotics or residues that can select for ARG resistance. For example, the efficacy of antibiotics wanes with increased ultraviolet light/sunlight exposure [63], but due to the ability of most metals to absorb photon energy with free electrons available, they will not break down when directly exposed to ultraviolet light [64]. Therefore, knowing that Superfund sites polluted hazardous waste containing heavy metals (including arsenic, cadmium, copper, and mercury) into the Salish Sea ecosystem in the past, it is not entirely surprising that we did not observe differences in HMRG carriage by proximity to Superfund sites; HMRGs may have spread throughout the Salish Sea's catchment area over the decades since the Superfund sites began their cleanup processes.

Since all study isolates carried HMRGs, a limitation of this study was the lack of a comparison group of genomes without HMRGs. The cross-sectional nature of the study precluded determination of the temporal relationship of the observed associations (e.g., did HMRG acquisition precede or follow ARG acquisition?), limiting any conclusions about causality. Similarly, we did not have access to exposure data (such as heavy metal measurements) from the environments where the samples were taken, and our reliance on proxy indicators (e.g., existence of a nearby Superfund site) may have resulted in misclassification of exposures, resulting in a bias toward the null hypothesis. Additionally, our sample contained only *E. coli* from animal and water sources, though it is possible that different HMRG patterns could be seen in isolates from human sources (which are the best characterized isolates). There was varying correlation between antibiotic resistance phenotype and carriage of corresponding known resistance genes, with 100% of *tet*-resistant/intermediate isolates carrying *tet* genes, but low correlation between aminoglycoside resistance phenotypes and genotypes. We also did not test for phenotypic heavy metal resistance; therefore, it is not clear whether the presence of HMRGs conferred actual resistance to their host *E. coli*. Lastly, the list of HMRGs we screened for was based on four heavy metals of public health relevance and was not an exhaustive screening of all HMRGs that could exist in the study samples. Genes conferring resistance to lead, often the most prevalent heavy metal contaminant in environments [2], were not found in our study. While the literature on heavy metal resistance is growing, characterization of

HMRGs in bacteria as markers for public health risk from heavy metal exposure has not been extensively explored recently.

The strengths of our study include the novel screening for HMRGs in *E. coli* genomes isolated from various sources in the Salish Sea ecosystem, adding to the characterization of potential health impacts of Superfund sites on nearby communities/ecosystems, using the environment and animals as sentinels for human health, utilization of a diversity of sample types (water and marine mammals), having a relatively large total sample size, and having exact base pair positions in the genome to determine linkage of ARGs and HMRGs.

Our study and past literature demonstrate evidence of widespread heavy metal resistance in *E. coli* isolates. The co-resistance of arsenic, cadmium, copper, and mercury-resistance genes with a diversity of resistance genes to commonly used antibiotics identified in our study have important potential implications for clinical applications. As *E. coli* continues to cycle through communities and the post-antibiotic era approaches, it is crucial to conduct global surveillance on the different resistance genes that *E. coli* isolates carry to recognize the limitations to future antibiotic efficacy. More specifically, understanding the specific genes that are present in the co-resistance of antibiotics and heavy metals in different reservoirs can help public health practitioners better predict what resistance patterns to expect in the future. Further, human, animal, and ecosystem aspects of AMR should be viewed in an integrative manner [65] that reflects the common environmental habitats and stressors of animals and humans through which resistance genes can be spread. Just as animals can serve as sentinels for human health, HMRGs surveillance may serve to estimate the nature and extent of heavy metal pollution in a given environment.

## Supporting information

**S1 Table. Total count of selected heavy metal resistance genes (arsenic, cadmium, copper, and mercury) in study isolates by Salish Sea region and source (N = 308).**
(DOCX)

**S2 Table. Co-resistance and linkage of heavy metal resistance genes (HMRGs), antibiotic resistance genes (ARGs), and plasmids in *E. coli* isolates by Salish Sea region and isolate sampling source (N = 25).**
(DOCX)

**S3 Table. Output from the Staramr tool with study *E. coli* genome contigs scanned against the Center for Genomic Epidemiology's ResFinder, PointFinder, and PlasmidFinder databases using the default 90% identity match and 60% minimum length coverage of the target gene thresholds, and 100% minimum total length coverage.**
(XLSX)

**S1 Fig. Total count of antibiotic resistance genes per *E. coli* isolate (N = 25).**
(TIF)

**S2 Fig. Total counts of heavy metal resistance genes and antibiotic resistance genes in *E. coli* isolates, harbor seals (N = 8).**
(TIF)

**S3 Fig. Total counts of heavy metal resistance genes and antibiotic resistance genes in *E. coli* isolates, river otters (N = 8).**
(TIF)

**S1 File. Supplementary results.**
(DOCX)

## Acknowledgments

We would like to thank Pauline Trinh, Evan Pepper, and David No for their bioinformatics assistance, Brian High for his coding help, Lauren Frisbie and Alexandria Vingino for their prior work on the study, and Cheryl Adler, Stephanie Norman, and Erica Fuhrmeister for their subject matter expertise regarding the manuscript.

## Author Contributions

**Conceptualization:** Ashley S. Tseng, Marilyn C. Roberts, Scott J. Weissman, Peter M. Rabinowitz.

**Data curation:** Ashley S. Tseng, Marilyn C. Roberts.

**Formal analysis:** Ashley S. Tseng.

**Funding acquisition:** Ashley S. Tseng, Peter M. Rabinowitz.

**Investigation:** Ashley S. Tseng, Marilyn C. Roberts, Peter M. Rabinowitz.

**Methodology:** Ashley S. Tseng, Marilyn C. Roberts, Scott J. Weissman, Peter M. Rabinowitz.

**Project administration:** Ashley S. Tseng, Marilyn C. Roberts, Peter M. Rabinowitz.

**Supervision:** Marilyn C. Roberts, Peter M. Rabinowitz.

**Validation:** Marilyn C. Roberts.

**Visualization:** Ashley S. Tseng.

**Writing – original draft:** Ashley S. Tseng.

**Writing – review & editing:** Ashley S. Tseng, Marilyn C. Roberts, Scott J. Weissman, Peter M. Rabinowitz.

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
