## [Decision Letter · Decision Letter 0]

19 Sep 2023

PONE-D-23-17575Carriage of arsenic, cadmium, copper, and mercury heavy metal resistance genes in *Escherichia coli* isolates from a marine ecosystem with a history of environmental pollutionPLOS ONE

Dear Dr. Rabinowitz,

Thank you for submitting your manuscript to PLOS ONE. After careful consideration, we feel that it has merit but does not fully meet PLOS ONE’s publication criteria as it currently stands. Therefore, we invite you to submit a revised version of the manuscript that addresses the points raised during the review process.

We look forward to receiving your revised manuscript.

Kind regards,

Ayman Elbehiry

Academic Editor

PLOS ONE

Reviewers' comments:

Reviewer's Responses to Questions

**Comments to the Author**

1. Is the manuscript technically sound, and do the data support the conclusions?

Reviewer #1: Partly

Reviewer #2: Yes

2. Has the statistical analysis been performed appropriately and rigorously? 

Reviewer #1: I Don't Know

Reviewer #2: Yes

3. Have the authors made all data underlying the findings in their manuscript fully available?

Reviewer #1: No

Reviewer #2: Yes

4. Is the manuscript presented in an intelligible fashion and written in standard English?

Reviewer #1: Yes

Reviewer #2: Yes

5. Review Comments to the Author

Reviewer #1: Peer review comments for minor revisions

The manuscript is based on impressive empirical evidence and makes an original contribution. Only minor revisions are needed before it can be published.

1. The authors conduct very relevant research, but unable to emphasis the relevance in the introduction.

2. To improve the readability of the paper, I suggest dividing the analysis into several subsections.

3. Table 2 is too long and difficult to understand, it should be adjusted.

4. Quality of figures should be improved, Text is very blurred and it is difficult to read black letters on dark green background.

5. There should be one more section conclusion/summary and future prospective to better reflect their work.

6. Line no. 23 page no. 2- sentence should be read as "We screened 18 functional . .

7. Line no. 24 page no. 2 sentence should be followed as "geographical variations . .

Reviewer #2: Comments to authors:

The current study is very interesting; however, the authors should address the following comments to improve the quality of the manuscript and to make it looks more beneficial to human health :

1- Title:

Please modify the title: All heavy metals mentioned in the title can be written between brackets in the end of the title, example: “study of heavy metal resistance genes in Escherichia coli isolates from a marine ecosystem with a history of environmental pollution, focusing on (arsenic, cadmium, copper, and mercury)”

2- Abstract:

- Rephrase the main conclusion of your findings to sound better.

3- Introduction: (it needs to be more informative), try to provide some information about the relationship between the effects of heavy metals in E.Coli and human health.

- Give a hint and some details about the spread of ARGs among and between humans and animals sources of HMRGs in bacterial environmental reservoirs.

- The authors should illustrate the importance of the study with regard to public health linked to the study objectives.

- Authors could add some paragraphs to fulfill the following gabs:

1- The relationship between the use of some metals as antibiotics for human treatment and antimicrobial resistance, heavy metal exposure, and resistance.

-Rephrase the aim of the work to be clear and better sound.

2- Write some paragraphs about Escherichia coli.

3- Material and methods

- R-write the methodology according to the PLOs guidelines.

-The Ethical statement should be placed before the sampling section since there samples taken from animals marine mammals (harbor seals, harbor porpoises, river otters).

-Isolation techniques analysis of samples isolates, should be modified.

- Methods of fecal samples collection and laboratory tests should be clearly described.

- Methods for isolating genes must be written, even if they are performed by contracting with a laboratory not associated with the study, regardless of its reliability or reputation

-Results:

- Increase the resolution of the two figures to be 600 dpi.

- Place Fig 1. “Total count of heavy metal resistance genes per E. coli isolate” and Fig.2 “Total count of antibiotic resistance genes per E. coli isolate source “ in their right place (not in the appendix)

- The manuscript is written well but needs minor revision for the grammatical errors and spelling mistakes.

- Please revise the references with the correct style of the journal.

6. PLOS authors have the option to publish the peer review history of their article (what does this mean?). If published, this will include your full peer review and any attached files.

Reviewer #1: **Yes: **Dr. Muhammad Abaid Ullah, DMMG, BZU, Multan, Pakistan

Reviewer #2: **Yes: **Dr Mustafa Mohammed Mustafa

---

## [Author Response · Author response to Decision Letter 0]

1 Nov 2023

Thank you for your review of our manuscript and the thoughtful comments. In line with your comments, we have revised the manuscript for greater clarity, readability, and phrasing in certain sections. Please refer to the "Response to Reviewers" document for detailed responses to your comments.

---

## [Editor Report · Decision Letter 1]

5 Nov 2023

Study of heavy metal resistance genes in *Escherichia coli* isolates from a marine ecosystem with a history of environmental pollution (arsenic, cadmium, copper, and mercury)

PONE-D-23-17575R1

Dear Dr. Peter Rabinowitz,

We’re pleased to inform you that your manuscript has been judged scientifically suitable for publication and will be formally accepted for publication once it meets all outstanding technical requirements.

Kind regards,

Ayman Elbehiry

Academic Editor

PLOS ONE
---

## [Editor Report · Acceptance letter]

9 Nov 2023

PONE-D-23-17575R1 

Study of heavy metal resistance genes in *Escherichia coli* isolates from a marine ecosystem with a history of environmental pollution (arsenic, cadmium, copper, and mercury) 

Dear Dr. Rabinowitz:

I'm pleased to inform you that your manuscript has been deemed suitable for publication in PLOS ONE. Congratulations! Your manuscript is now with our production department. 

Kind regards, 

on behalf of

Professor Ayman Elbehiry 

Academic Editor

PLOS ONE